# Identification of Coronary Morphological Damage in Patients with Chronic Inflammatory Rheumatic Diseases

**DOI:** 10.3390/diagnostics15070922

**Published:** 2025-04-02

**Authors:** Elena Heras-Recuero, Juan Antonio Martínez-López, Macarena Garbayo-Bugeda, Álvaro Castrillo-Capilla, Teresa Blázquez-Sánchez, Arantxa Torres-Roselló, Antia García-Fernández, Javier Llorca, Raquel Largo, Juan Antonio Franco-Peláez, José Tuñón, Miguel Ángel González-Gay

**Affiliations:** 1Division of Rheumatology, Fundación Jiménez Díaz, 28040 Madrid, Spain; elena.herasr@fjd.es (E.H.-R.); jamartinez@quironsalud.es (J.A.M.-L.); teresa.blazquez@quironsalud.es (T.B.-S.); arantxa.torres@quironsalud.es (A.T.-R.); antia.garcia@quironsalud.es (A.G.-F.); rlargo@fjd.es (R.L.); 2Instituto de Investigación Sanitaria (IIS)-Fundación Jiménez Díaz, 28040 Madrid, Spain; 3Department of Cardiology, Instituto de Investigación Sanitaria (IIS)-Fundación Jiménez Díaz, 28040 Madrid, Spain; macarena.garbayo@quironsalud.es (M.G.-B.); alvaro.castrillo@quironsalud.es (Á.C.-C.); jafranco@quironsalud.es (J.A.F.-P.); jtunon@quironsalud.es (J.T.); 4CIBER Epidemiología y Salud Pública (CIBERESP), Department of Medical and Surgical Sciences, University of Cantabria, 39011 Santander, Spain; javierllorca1958@gmail.com; 5Renal, Vascular and Diabetes Research Laboratory, IIS-Fundación Jiménez Díaz, 28040 Madrid, Spain; 6Faculty of Medicine, Universidad Autónoma de Madrid, 28049 Madrid, Spain; 7Centro de Investigación en Red de Enfermedades Cardiovasculares (CIBERCV), 28029 Madrid, Spain; 8Medicine and Psychiatry Department, University of Cantabria, 39005 Santander, Spain

**Keywords:** chronic inflammatory rheumatic diseases, coronary heart disease, coronary angiography

## Abstract

**Objective:** Patients with chronic inflammatory rheumatic diseases (CIRDs) have a higher incidence of coronary artery disease (CAD) due to accelerated atherogenesis. This study aimed to assess the extent and location of CAD lesions in CIRD patients compared to non-CIRD patients. **Methods:** A retrospective study was conducted on CIRD patients (rheumatoid arthritis, ankylosing spondylitis, and psoriatic arthritis) who underwent coronary angiography at Hospital Fundación Jiménez Díaz (Madrid, Spain) between 2018 and 2022. For each CIRD patient, at least two frequency-matched controls were selected based on sex, age (±2 years), diabetic status, and clinical indication for coronary angiography. The indications for coronary angiography in both groups were chronic coronary syndrome and acute coronary syndrome with or without ST elevation. **Results:** A total of 66 CIRD patients were included, with 42 (63.6%) women, and a median age of 66.6 years (range: 58.3–75.2). Compared to the controls, CIRD patients had a higher number of affected coronary arteries (2.03 vs. 1.56, *p* = 0.03). The mid-anterior descending artery and the right posterior descending artery were more frequently involved in CIRD patients than in controls (odds ratio [OR] of 2.45 and 3.53, respectively, *p* ≤ 0.02 for both comparisons). The frequency of coronary calcification was higher in CIRD patients, though the difference did not reach statistical significance (5 of 66 in CIRD patients vs. 3 of 140 in non-CIRD controls, OR of 3.74, *p* = 0.06). Revascularization was more commonly performed in patients with CIRD (50 of 66 vs. 85 of 140 in those without CIRD (OR: 2.02 [95% CI: 1.01–4.18]; *p* = 0.03). **Conclusions:** Patients with CIRD exhibit more extensive CAD, with a higher propensity for involvement inthe mid-anterior descending and right posterior descending arteries compared to patients without CIRD. These findings highlight the need for closer cardiovascular monitoring and early risk stratification in CIRD patients to improve the detection and management of CAD.

## 1. Introduction

Chronic inflammatory rheumatic diseases (CIRDs) are characterized by persistent inflammation and immune system dysregulation, which often lead to an increased risk of cardiovascular disease [1,2,3]. Within the category of CIRD, we include conditions such as rheumatoid arthritis [4,5,6,7], ankylosing spondylitis [8,9,10,11], and psoriatic arthritis [12,13,14], which have been associated with accelerated atherosclerosis and an increased risk of mortality due to cardiovascular events.

CIRD, such as rheumatoid arthritis, and atherosclerosis share common inflammatory pathways [15]. Patients with CIRD often experience endothelial dysfunction [16,17,18,19,20], which is an early stage in the development of atherogenesis. Endothelial dysfunction due to arterial wall damage leads to increased permeability. In addition, circulating low-density lipoproteins (LDLs) are depositedin the intima of the damaged endothelial vascular wall. LDL becomes oxidized, triggering inflammation and attracting macrophages into the intima. Macrophages engulf the oxidized LDL, forming foam cells. As the foam cells replicate and release cytokines, a fatty streak develops, promoting the migration of smooth muscle cells into the intima. These smooth muscle cells release collagen and proteoglycans, leading to the formation of an atherosclerotic plaque. The plaque develops a fibrous cap, which can rupture, causing thrombosis. Additionally, over time, the plaque may become calcified, leading to arterial stenosis and ischemia [1].

A higher frequency of subclinical atherosclerosis has also been observed in CIRD, as demonstrated by imaging studies such as carotid ultrasound [21,22,23,24,25,26], which frequently detects atheromatous plaques in these patients, and cardiac computed tomography (CT), used for coronary calcium quantification [27,28]. In this regard, both carotid ultrasound and coronary CT have been identified as useful surrogate markers of coronary artery disease (CAD) in the general population as well as in CIRD patients [27].

One unresolved issue is whether the severity of CAD in CIRD patients is comparable to that in individuals without underlying autoimmune or chronic inflammatory diseases. Specifically, it is important to assess whether the extent of CAD differs in terms of the number and location of affected coronary vessels when comparing CIRD patients to those with CAD but no inflammatory or autoimmune conditions. To address this question, we conducted a retrospective study of CIRD patients, including those with rheumatoid arthritis, ankylosing spondylitis, and psoriatic arthritis, who were treated at the Rheumatology Department of a reference hospital in central Spain and underwent coronary angiography. In this regard, while it is well established that patients with CIRD have an increased risk of developing CAD, the extent and anatomical distribution of CAD in this population remain inadequately characterized. Most studies focus on general cardiovascular risk in CIRD patients but do not provide detailed assessments of coronary involvement compared to individuals without inflammatory or autoimmune diseases. This knowledge gap limits the ability to develop tailored screening and management strategies for these patients.

We hypothesize that CIRD patients exhibit more extensive CAD, with distinct patterns of coronary involvement compared to non-CIRD individuals. Specifically, we expect to find a higher burden of atherosclerotic disease in certain coronary segments. This study aims to address this gap by analyzing coronary angiography findings in CIRD patients, providing insights that could improve cardiovascular risk stratification and targeted interventions.

## 2. Patients and Methods

This study included all CIRD patients treated at the Rheumatology Division of Hospital Fundación Jiménez Díaz in Madrid, Spain, who underwent coronary angiography between 2018 and 2022. They included patients with rheumatoid arthritis, ankylosing spondylitis, and psoriatic arthritis who fulfilled the classification criteria for the diagnosis of each CIRD [29,30,31].

For each CIRD patient, at least two frequency-matched controls were selected based on sex, age (±2 years), diabetic status (yes/no), and clinical indication for coronary angiography. The controls had no inflammatory rheumatic diseases or other inflammatory or malignant conditions.

The indications for coronary angiography were applicable to both patients with CIRD and non-rheumatic patients. These included chronic coronary syndrome, acute coronary syndrome without ST elevation, and acute coronary syndrome with ST elevation.

The study procedures were conducted in accordance with the Helsinki Declaration of 1975, as revised in 2000. Although it was a retrospective study, ethical committee approval was obtained from the Ethical Committee of Fundación Jiménez Díaz, Madrid, Spain (PIC 071-23_FJD)-07-07-2023.

### Statistical Analysis

Patient demographics were presented as mean and standard deviation (SD), or median and interquartile range [IQR] for skewed quantitative variables, and as absolute numbers and relative frequencies (%) for qualitative variables. Categorical variables were compared using Fisher’s exact test. Continuous variables were compared using Student’s *t*-test.

To measure the association between CAD extension and CIRD, we estimated odds ratios (ORs) with 95% confidence interval (CI). ORs were not adjusted for age and gender as the control sample was frequency-matched for these two variables. We also estimated non-parametric area under the ROC curve (AUROC). OR > 1 or AUROC > 0.5 indicate that specific CAD was more frequent in patients with CIRD, while OR < 1 or AUROC < 0.5 indicate the opposite.

All statistical analyses were carried out with the package Stata 18/SE.

## 3. Results

Sixty-six patients with CIRD were assessed. A total of 41 were diagnosed with rheumatoid arthritis, 14 with ankylosing spondylitis, and 11 with psoriatic arthritis.

Additionally, 42 of the 66 patients with CIRD were women (63.6%), with a median age of 66.6 years (range: 58.3–75.2) at the time of the coronary event.

Moreover, 68 percent of patients with CIRD had a history of hypertension, 23% of diabetes, and 82% of dyslipidemia. At the time of the study, the mean values of total cholesterol and LDL-cholesterol were 180 ± 47 mg/dL and 105 ± 39 mg/dL, respectively. Sixty percent had a smoking history, and thirty-six percent were current smokers at the time of the CAD event. At that time, 35% had a body mass index (BMI) ≥25. The median (IQ range) duration of the disease from the disease diagnosis at the time of the CAD event was 9 (IQ range: 3.4–12.7) years. At the time of the CAD event, 14 (21%) of the 66 CIRD patients had received or were undergoing biologic therapy, and nearly 60% of those with rheumatoid arthritis were being treated with methotrexate.

When compared with matched controls, no differences in age or gender were found between CIRD patients and those without CIRD who underwent coronary angiography (Table 1). However, the number of affected coronary arteries was higher in the CIRD group compared to the controls (2.03 vs. 1.56; *p*= 0.03).

Table 2 shows the distribution of CAD damage in patients with and without CIRD. Regarding CAD distribution, the mid anterior descending artery and right posterior descending artery were more commonly affected in the CIRD group than in controls (OR: 2.45 [95% CI: 1.29–4.65] and 3.53 [95% CI: 1.06–12.5]), respectively; *p* ≤ 0.02 for both comparisons). In this regard, the coronary angiographic damage in both arteries in CIRD patients showed OR values greater than 1, indicating that atherosclerotic damage in these arteries was more frequent in CIRD patients than in the control group of individuals without CIRD who underwent coronary angiography. Moreover, the area under the ROC curve for atherosclerotic damage in both arteries in CIRD patients was greater than 0.5, supporting an association with atherosclerotic disease in these arteries in CIRD patients compared to those without CIRD (Table 2).

The frequency of coronary calcification was higher in patients with CIRD than in controls. However, the difference did not reach statistical significance (5 of 66 in CIRD patients vs. 3 of 140 in non-CIRD controls; OR: 3.74; *p*= 0.06) (Table 3).

No other significant differences in the frequency of arterial coronary damage between patients with and without CIRD were observed (Table 3).

A forest plot showing odds ratios and 95% confidence intervals (CIs) for each artery studied is shown in Figure 1.

Interestingly, revascularization (angioplasty or coronary artery bypass surgery) was more commonly performed in patients with coronary artery disease and CIRD than in those without CIRD (Table 3). In the group of patients with CIRD, 50 of 66 underwent revascularization, compared to 85 of 140 in those without CIRD (OR: 2.02 [95% CI: 1.01–4.18]; *p* = 0.03).

In a subsequent step, we compared whether coronary damage following coronary angiography assessment differed between patients with rheumatoid arthritis, ankylosing spondylitis, or psoriatic arthritis (Table 4).

In most cases, there were no significant differences in the frequency of arterial damage detected by coronary angiography among the three CIRD groups. In this regard, the only significant difference was found in the involvement of the mid-distal circumflex artery, which was absent in patients with rheumatoid arthritis (Table 4). However, these results should be interpreted with caution due to the small number of patients with ankylosing spondylitis and psoriatic arthritis.

## 4. Discussion

Patients with CIRD, such as rheumatoid, ankylosing spondylitis, and psoriatic arthritis, are at an increased risk of accelerated atherosclerosis and cardiovascular events due to the chronic inflammation characteristic of these conditions [1]. Genetic factors have also been proposed to influence the risk of cardiovascular disease in these patients [32]. In the present study, we investigated the distribution of CAD in CIRD patients. Our findings show that these patients exhibit a higher number of affected coronary arteries compared to matched controls, supporting the hypothesis that chronic inflammation contributes to more extensive coronary atherosclerosis. Notably, we observedagreater involvement of the mid-anterior descending and right posterior descending arteries in CIRD patients. However, these results are of borderline significance. Therefore, this higher frequency of involvement of these arteries in patients with CIRD compared to non-CIRD patients deserves further replication in an unrelated population.

While information on the location and extent of coronary damage in CIRD patients is limited, several studies have attempted to explore this issue. With respect to this, Aubry et al. analyzed the differences in CAD between individuals with and without rheumatoid arthritis [33]. They conducted a retrospective analysis, based on autopsy data from patients who died between 1980 and 2004, which included 157 patients with rheumatoid arthritis and 173 control patients without CIRD. The study revealed that patients with rheumatoid arthritis had a significantly higher burden of atherosclerosis, including more severe coronary artery disease and a greater frequency of multi-vessel involvement. Additionally, patients with rheumatoid arthritis exhibited more advanced forms of CAD, despite being younger on average than the control group [33]. Holmqvist et al. performed a population-based study using nationwide clinical, health, and demographic registers in Sweden. They compared 2985 patients with rheumatoid arthritis and 10,290 non-rheumatoid arthritis patients who underwent coronary angiography between 2006 and 2015 [34]. However, their study found no significant differences in the presence and distribution of clinically significant coronary stenoses between the two groups, regardless of disease duration, sex, or cardiovascular risk factors [34].

A systematic review and meta-analysis, however, provided evidence of an increased prevalence of asymptomatic CAD in patients with rheumatoid arthritis, measured using coronary calcium scores and coronary CT angiography. The analysis of eight studies (788 rheumatoid arthritis patients and 1641 controls) revealed that patients with rheumatoid arthritis had a significantly higher risk of coronary artery disease and higher coronary calcium scores compared to controls [35]. Patients with rheumatoid arthritis also had a higher prevalence of moderate-to-severe coronary artery disease and multivessel disease, with disease duration and activity linked to higher coronary calcium scores and the presence of high-risk coronary plaques [35]. Additionally, studies on ankylosing spondylitis patients found that atherosclerotic plaques in the coronary arteries were significantly more prevalent in these individuals compared to controls [36]. In our own cohort of CIRD patients, we also observed an increased frequency of coronary calcification, though the difference did not reach statistical significance, potentially due to our sample size.

Further supporting our findings, Karpouzas et al. assessed the presence, burden, and composition of coronary plaque in patients with rheumatoid arthritis who were asymptomatic or had no clinical diagnosis of CAD [37]. They compared 150 rheumatoid arthritis patients with 150 matched controls using 64-slice CT angiography. Their study found that a higher proportion of rheumatoid arthritis patients had coronary plaque compared to controls. Additionally, these patients had more multivessel disease, both non-obstructive and obstructive, as well as higher stenosis severity and plaque extent, even after adjusting for cardiac risk factors [37]. This result is consistent with our finding of a higher frequency of revascularization procedures (angioplasty or coronary artery bypass surgery) in CIRD patients compared to controls, likely due to the more severe coronary damage present even in the preclinical stages of coronary artery disease.

One possible explanation for increased atherosclerotic coronary findings in patients with CIRD is the persistence of chronic inflammation, which has been associated with the development of subclinical atherosclerosis. In rheumatoid arthritis, for example, the magnitude and chronicity of the inflammatory response, as measured by C-reactive protein, have been linked to the progression of atherosclerosis [38,39]. In addition, the duration of the disease is also an important factor implicated in the development of atherosclerosis in patients with CIRD. In this regard, a study on patients with rheumatoid arthritis without clinically evident cardiovascular disease found that those with carotid plaques detected via ultrasound assessment had a significantly longer disease duration than those without plaques [21]. These findings highlight the importance of monitoring coronary health in CIRD patients, even in the absence of clinical CAD, as they are at higher risk of cardiovascular events due to the cumulative effects of chronic inflammation on the coronary vasculature. In addition to the tight control of disease activity to minimize the inflammatory burden, the strict management of traditional cardiovascular risk factors is essential for patients with CIRD. In this context, Karpouzas et al. provided valuable insights into the relationship between statin use and coronary atherosclerosis progression in RA [40]. These authors emphasized the complex interplay between chronic inflammation in RA and increased cardiovascular disease risk, highlighting that statins not only help manage lipid levels but may also slow the progression of coronary atherosclerosis. Given the increased risk of heart disease in these patients, the long-term cardiovascular benefits of statins are particularly crucial in the management of CIRD. These findings highlight the importance of incorporating statin therapy alongside disease-modifying treatments to reduce cardiovascular risk in CIRD patients.

An aspect not evaluated in our study was the implication of coronary microvascular dysfunction (CMD) in patients with CIRD. In this regard, Faccini et al. assessed the relationship between CIRD and CMD [41]. The authors highlighted how systemic inflammation in conditions such as rheumatoid arthritis and systemic lupus erythematosus contributes to CMD, independent of traditional cardiovascular risk factors. According to these authors, the underlying mechanism, including endothelial dysfunction, immune system activation, and oxidative stress, lead to impaired coronary microcirculation. They emphasized the clinical relevance of CMD in these patients, as it increases the risk of cardiovascular events and warrants early detection and management [41]. Interestingly, more recently, Geng et al. highlighted the role of the Index of Microcirculatory Resistance (IMR), a key parameter for assessing coronary microvascular function [42]. The authors discussed how IMR can aid in identifying microvascular dysfunction, predicting patient outcomes, and guiding treatment strategies. This procedure may be applicable to patients with CIRD [42].

The EULAR (European League Against Rheumatism) 2015/2016 recommendations update guidance on managing CVD risk in patients with rheumatoid arthritis and other CIRD emphasized the need of early detection and intervention, as these patients have a higher risk of CVD. In addition to using traditional CVD risk scores (such as SCORE and Framingham), which should be adjusted by a 1.5 multiplication factor in RA due to risk underestimation, the recommendations highlighted the importance of smoking cessation, weight control, and promoting physical activity. According to these recommendations, hypertension, diabetes, and lipid levels should be managed similarly to the general population. Furthermore, the recommendations emphasize the need for controlling systemic inflammation using disease-modifying antirheumatic drugs such as methotrexate, tumor necrosis factor inhibitors, and interleukin-6 blockers to reduce cardiovascular risk [43].

Regarding lipid management, these recommendations indicate that statins should be considered for patients at high CVD risk, with treatment targets similar to those in the general population [43]. However, an unanswered question remains: should there be a standardized LDL-cholesterol target for initiating statin therapy in patients with CIRD, similar to the target used in the cardiovascular risk classification for patients with diabetes? In individuals with diabetes alone, the LDL-cholesterol target is 70 mg/dL, while in those with established cardiovascular disease, it is 55 mg/dL. Notably, many patients with CIRD have carotid plaques when assessed via carotid ultrasound. This finding places these patients in the category of very high cardiovascular risk. According to the 2019 ESC/EAS guidelines for the management of dyslipidemia, patients at very high cardiovascular risk should aim for an LDL-cholesterol reduction of ≥50% from baseline, with an LDL-cholesterol goal of <55 mg/dL [44,45]. Moreover, since CIRD itself is considered a condition associated with high cardiovascular risk, as indicated in the 2019 ESC/EAS guidelines for the management of dyslipidemia [36,37], we feel that these patients should be treated with an LDL-cholesterol goal of <70 mg/dL.

Our study has a number of potential limitations that we are pleased to address. First, this is a retrospective study. In addition, the group of CIRD patients was heterogeneous, as it encompassed three different conditions, and the number of patients for each of the three CIRD types was small. However, various studies have demonstrated that rheumatoid arthritis, ankylosing spondylitis, and psoriatic arthritis are each independently associated with accelerated atherosclerosis and an increased risk of CAD. This is why we included patients with these three conditions as a single group. However, it has several strengths. In this regard, most studies focus on general cardiovascular risk in CIRD patients but do not provide detailed assessments of coronary involvement compared to individuals without inflammatory or autoimmune diseases. In this study, we specifically describe the pattern of coronary involvement in patients with CIRD and compare it with that of non-CIRD individuals. This comparison has been scarcely addressed in the literature. Furthermore, we observed a higher burden of atherosclerotic disease in certain coronary segments. This information could be valuable for improving cardiovascular risk stratification and guiding targeted interventions in patients with CIRD.

Regarding future insights, our study shows that patients with CIRD have more extensive coronary artery involvement than non-CIRD individuals. This highlights the importance of regular cardiovascular monitoring for CIRD patients, especially given their increased risk of more severe CAD. These findings support the need for longitudinal studies to explore the progression of CAD in CIRD patients over time, assessing how early intervention and changes in inflammatory markers correlate with improvements or worsening of cardiovascular health. Moreover, the involvement of specific coronary arteries (mid-anterior descending artery and right posterior descending artery) was higher in our CIRD patients. This could be important for understanding disease-specific patterns of atherosclerosis. Based on this, further studies could investigate the underlying mechanisms contributing to this preferential arterial involvement. In addition, revascularization was more common in our CIRD patients than in non-CIRD individuals, which could indicate that they experience more severe or widespread CAD, leading to higher rates of procedural intervention. Therefore, a follow-up study could examine the outcomes of revascularization in CIRD patients compared to non-CIRD patients.

## 5. Conclusions

Patients with CIRD have more widespread CAD and a greater tendency for involvement of the mid-anterior descending and right posterior descending arteries, compared to those without CIRD.

## Figures and Tables

**Figure 1 diagnostics-15-00922-f001:**
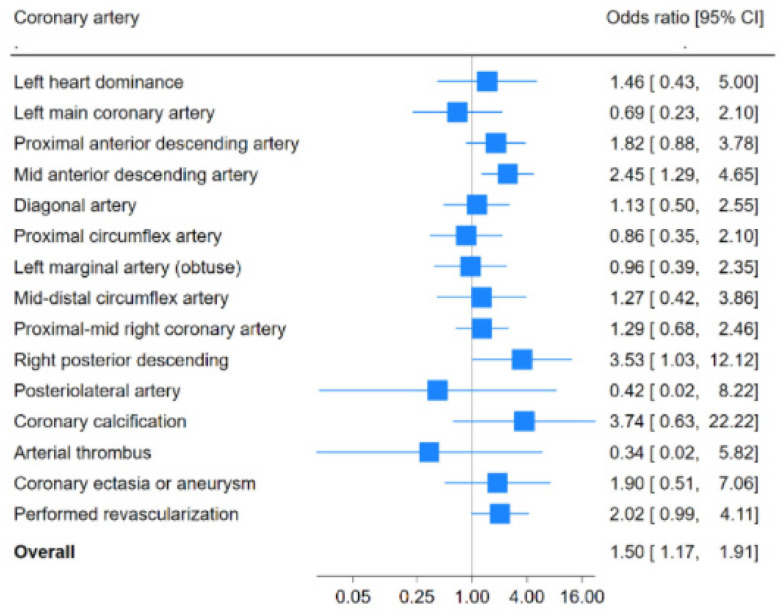
Forest plot displaying odds ratios and 95% confidence intervals (CIs) for each artery studied in Table 2 and Table 3.

**Table 1 diagnostics-15-00922-t001:** Age, sex, and number of affected coronary arteries in patients with chronic inflammatory rheumatic diseases (CIRD) and those controls without CIRD.

Variable	CIRD	Non-CIRD	*p*
Age (mean ± SD)	66.9 ± 11.2	66.8 ± 11.5	0.95
Age (median, IQR)	66.6 (58.3, 75.2)	66.3 (58.1, 75.8)	
Gender (men/women)	42 (63.6%)/24	89 (63.6%)/51	0.99
**Number of coronary arteries affected**	**2.03 ± 1.32**	**1.56 ± 1.45**	**0.03**

SD: Standard deviation. IQR: Interquartile range. Data **in bold** indicate significant differences between CIRD patients and controls.

**Table 2 diagnostics-15-00922-t002:** Coronarography differences between patients with chronic inflammatory rheumatic disease (CIRD) and matched controls without CIRD (non-CIRD).

Coronary Artery	CIRD *	Non-CIRD **	Odds Ratio (95% CI)	*p*	Area Under the ROC Curve (95% CI)
Left heart dominance	6/66	9/140	1.46 (0.41, 4.81)	0.49	0.514 (0.473, 0.554)
Left main coronary artery	6/65	18/140	0.69 (0.21, 1.94)	0.45	0.482 (0.437, 0.527)
Proximal anterior Descending artery	20/66	27/140	1.82 (0.87, 3.75)	0.08	0.555 (0.490, 0.620)
**Mid anterior** **descending artery**	**37/66**	**48/140**	**2.45 (1.29, 4.65)**	**0.003**	**0.609 (0.537, 0.681)**
Diagonal artery	13/66	25/140	1.13 (0.49, 2.50)	0.75	0.509 (0.451, 0.567)
Proximal circumflex artery	10/66	24/140	0.86 (0.34 2.03)	0.72	0.490 (0.436, 0.544)
Left marginal artery (obtuse)	10/66	22/140	0.96 (0.38, 2.28)	0.92	0.497 (0.444, 0.550)
Mid-distal circumflex artery	7/66	12/140	1.27 (0.40, 3.69)	0.64	0.510 (0.466, 0.554)
Proximal-mid right coronary artery	27/66	49/140	1.29 (0.67, 2.44)	0.41	0.530 (0.458, 0.601)
**Right posterior descending**	**9/66**	**6/140**	**3.53 (1.06, 12.5)**	**0.02**	**0.547 (0.502, 0.592)**
Posterolateral artery	1/66	5/140	0.42 (0.01, 3.83)	0.41	0.490 (0.468, 0.511)

** n* with each coronary artery affected/total number of patients with CIRD. ** *n* with each coronary artery affected/total number of patients without CIRD.

**Table 3 diagnostics-15-00922-t003:** Morphological coronary differences between patients with CIRD and matched controls without CIRD.

	CIRD *	Non-CIRD **	Odds Ratio (95% CI)	*p*	Area Under the ROC Curve (95% CI)
Coronary calcification	5/66	3/140	3.74 (0.70, 24.7)	0.06	0.527 (0.493, 0.562)
Arterial thrombus	1/66	6/140	0.34 (0.01, 2.93)	0.31	0.486 (0.464, 0.509)
Coronary ectasia or aneurysm	6/66	7/140	1.90 (0.50, 6.90)	0.26	0.521 (0.481, 0.560)
**Performed Revascularization**	**50/66**	**85/140**	**2.02 (1.01, 4.18)**	**0.03**	**0.587 (0.517, 0.657)**
No-reflow	2/66	0/140	---	---	0.515 (0.494, 0.536)

** n* with each coronary artery affected/total number of patients with CIRD. ** *n* with each coronary artery affected/total number of patients without CIRD. Data **in bold** indicate significant differences between CIRD patients and controls.

**Table 4 diagnostics-15-00922-t004:** Coronarography differences among patients with rheumatoid arthritis, ankylosing spondylitis, and psoriatic arthritis.

Coronary Artery	Rheumatoid Arthritis *	Ankylosing Spondylitis *	Psoriatic Arthritis *	*p* **
Left heart dominance	4/41	1/14	0/11	0.35
Left main coronary artery	4/41	0/14	2/11	0.30
Proximal anterior Descending artery	16/41	2/14	2/11	0.15
Mid anterior descending artery	25/41	7/14	5/11	0.60
Diagonal artery	8/41	1/14	4/11	0.20
Proximal circumflex artery	8/41	1/14	1/11	0.55
Left marginal artery (obtuse)	8/41	1/14	1/11	0.55
**Mid-distal circumflex artery**	**0/41**	**3/14**	**4/11**	**<0.001**
Proximal-mid right coronary artery	16/41	5/14	6/11	0.64
Right posterior descending	4/41	2/14	3/11	0.31
Posterolateral artery	0/41	1/14	0/11	0.38
Coronary calcification	4/41	0/14	1/11	0.66
Arterial thrombus	0/41	0/14	1/11	0.17
Coronary ectasia or aneurysm	3/41	2/14	1/11	0.83
Performed Revascularization	31/41	10/14	9/11	0.89
No-reflow	2/41	0/14	0/11	1.00

** n* with each coronary artery affected/total number of patients with each CIRD. ** *p* values estimated with Fisher exact test. Data **in bold** indicate significant differences between CIRD patients and controls.

## Data Availability

The data sets used and/or analyzed in the present study are available from the corresponding author upon request.

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
