# Peer review of "Identification of Coronary Morphological Damage in Patients with Chronic Inflammatory Rheumatic Diseases"

_diagnostics, 2025, doi:10.3390/diagnostics15070922_

Round 1

Reviewer 1 Report

Comments and Suggestions for Authors

The authors presented a brief report examining the morphological damage of coronary artery disease (CAD) in chronic inflammatory rheumatic diseases (CIRD). This topic has been widely studied, as the authors also mentioned in the discussion section, but it provides new information in terms of specific lesion distribution in coronary arteries in CIRD. After reviewing the manuscript, I would like to make a few suggestions for improving the paper.

1. In the conclusion of abstract, state what the results of the study provide in terms of clinical applications.

2. At the end of the introduction, state your hypothesis and the gap in the literature that the study aims to fill more clearly.

3. In the methods section, state the patient selection and exclusion criteria more clearly. For example, how did you prevent bias in the selection of patients in the control group? There is heterogeneity in terms of disease groups among CIRD patients. The sample size is insufficient for subgroup analyses of these diseases. What were the participants' HT and smoking status? It is important to know the disease activity status, disease duration and treatments of CIRD patients. Especially the duration of the disease is an important effect on the progression of inflammatory processes.

4. Some results have borderline significance, carefully emphasize this in the discussion section.

5. Limitations of the study and clinical practical recommendations are missing. Indicate the strengths and weaknesses of your study and provide a suggestion and insight for future studies.

6. In some places, "CRID" is written instead of "CIRD".

7. As a reviewer, the most important question that arose after reading this study is: Should there be a standardized LDL-C target for initiating statin therapy in patients with CIRD, similar to the cardiovascular risk classification used for patients with diabetes? For instance, in individuals with diabetes alone, the LDL-C target is 70 mg/dL, while in those with established cardiovascular disease, it is 55 mg/dL. Is there an equivalent target value for patients with CIRD, or should there be one? What does the existing literature suggest on this matter, and what are the authors' recommendations? Please discuss this in the Discussion section.

Comments on the Quality of English Language

The English language is generally understandable, but some typos need correction.

Author Response

Reviewer 1

Comments and Suggestions for Authors

The authors presented a brief report examining the morphological damage of coronary artery disease (CAD) in chronic inflammatory rheumatic diseases (CIRD). This topic has been widely studied, as the authors also mentioned in the discussion section, but it provides new information in terms of specific lesion distribution in coronary arteries in CIRD. After reviewing the manuscript, I would like to make a few suggestions for improving the paper.

  1. In the conclusion of abstract, state what the results of the study provide in terms of clinical applications.

RESPONSE:

We have modified the conclusion of the abstract as suggested by the Reviewer:

Conclusion: Patients with CIRD exhibit more extensive CAD, with a higher propensity for involvement of the mid-anterior descending and right posterior descending arteries compared to patients without CIRD. These findings highlight the need for closer cardiovascular monitoring and early risk stratification in CIRD patients to improve detection and management of CAD.

  1. At the end of the introduction, state your hypothesis and the gap in the literature that the study aims to fill more clearly.

RESPONSE: We greatly appreciate the accurate comment raised by the Reviewer, which clearly strengthens the rationale for our study.

In this regard, while it is well established that patients with CIRD have an increased risk of developing CAD, the extent and anatomical distribution of CAD in this population remain inadequately characterized. Most studies focus on general cardiovascular risk in CIRD patients but do not provide detailed assessments of coronary involvement compared to individuals without inflammatory or autoimmune diseases. This knowledge gap limits the ability to develop tailored screening and management strategies for these patients.

We hypothesize that CIRD patients exhibit more extensive CAD, with distinct patterns of coronary involvement compared to non-CIRD individuals. Specifically, we expect to find a higher burden of atherosclerotic disease in certain coronary segments. This study aims to address this gap by analyzing coronary angiography findings in CIRD patients, providing insights that could improve cardiovascular risk stratification and targeted interventions.

  1. In the methods section, state the patient selection and exclusion criteria more clearly. For example, how did you prevent bias in the selection of patients in the control group?

Response: To prevent bias in the selection of CIRD patients we selected all CIRD who underwent coronary angiography between 2018 and 2022.

For each CIRD patient, at least two frequency-matched controls were selected based on sex, age (+/- 2 years), diabetic status (yes/no), and clinical indication for coronary angi-ography. The controls had no inflammatory rheumatic diseases or other inflammatory or malignant conditions.

There is heterogeneity in terms of disease groups among CIRD patients. The sample size is insufficient for subgroup analyses of these diseases.

Response: We agree with the Reviewer that the group of CIRD patients was heterogeneous, as it encompassed several CIRD, and the number of patients for each of the three CIRD types was small. However, various studies have demonstrated that rheumatoid arthritis, ankylosing spondylitis, and psoriatic arthritis are each independently associated with accelerated atherosclerosis and an increased risk of CAD. This is why we included patients with these three conditions as a single group.

We have included this point in the Discussion section as a potential limitation as follows:

Our study has a number of potential limitations that we are pleased to address. First, this is a retrospective study. In addition, the group of CIRD patients was heterogeneous, as it encompassed three different conditions, and the number of patients for each of the three CIRD types was small. However, various studies have demonstrated that rheumatoid arthritis, ankylosing spondylitis, and psoriatic arthritis are each independently associated with accelerated atherosclerosis and an increased risk of CAD. This is why we included patients with these three conditions as a single group.

What were the participants' HT and smoking status? It is important to know the disease activity status, disease duration and treatments of CIRD patients.

Especially the duration of the disease is an important effect on the progression of inflammatory processes.

Response: Since we retrospectively evaluated the patients' clinical records, we are pleased to include the available data in the Results section:

Sixty-eight percent of patients with CIRD had a history of hypertension, 23% of dia-betes, and 82% of dyslipidemia. At the time of the study the mean values of total cholesterol and LDL-cholesterol were 180±47 and 105±39 mg/dl, respectively. Sixty percent had a smoking history, and 36% were current smokers at the time of the CAD event. At that time, 35% had a body mass index (BMI) ≥25. The median (IQ range) duration of the disease from the disease diagnosis at the time of the CAD event was 9 (IQ range: 3.4-12.7) years. At the time of the CAD event, 14 (21%) of the 66 CIRD patients had received or were undergoing biologic therapy, and nearly 60% of those with rheumatoid arthritis were being treated with methotrexate.

We have also highlighted the importance of disease duration as a relevant factor in the development of accelerated atherosclerosis in patients with CIRD in the Discussion section, as follows:

In addition, the duration of the disease is also an important factor implicated in the development of atherosclerosis in patients with CIRD. In this regard, a study on patients with rheumatoid arthritis without clinically evident cardiovascular disease found that those with carotid plaques detected via ultrasound assessment had a significantly longer disease duration than those without plaques [21].

  1. Some results have borderline significance, carefully emphasize this in the discussion section.

Response: As indicated by the Reviewer, in the Discussion we have emphasized that some results have borderline significance. 

At the end of the first paragraph, we have added the following:

However, these results are of borderline significance. Therefore, this higher frequency of involvement of these arteries in patients with CIRD compared to non-CIRD patients deserves further replication in an unrelated population.

  1. Limitations of the study and clinical practical recommendations are missing. Indicate the strengths and weaknesses of your study and provide a suggestion and insight for future studies.

Response: As indication by the Reviewer, in the last paragraph of the Discussion section before conclusion s we have added the following:

Our study has a number of potential limitations that we are pleased to address. First, this is a retrospective study. In addition, the group of CIRD patients was heterogeneous, as it encompassed three different conditions, and the number of patients for each of the three CIRD types was small. However, various studies have demonstrated that rheumatoid arthritis, ankylosing spondylitis, and psoriatic arthritis are each independently associated with accelerated atherosclerosis and an increased risk of CAD. This is why we included patients with these three conditions as a single group. However, it has several strengths. In this regard, most studies focus on general cardiovascular risk in CIRD patients but do not provide detailed assessments of coronary involvement compared to individuals without inflammatory or autoimmune diseases. In this study, we specifically describe the pattern of coronary involvement in patients with CIRD and compare it with that of non-CIRD individuals. This comparison has been scarcely addressed in the literature. Furthermore, we observed a higher burden of atherosclerotic disease in certain coronary segments. This information could be valuable for improving cardiovascular risk stratification and guiding targeted interventions in patients with CIRD.

                Regarding future insights, our study shows that patients with CIRD have more extensive coronary artery involvement than non-CIRD individuals. This highlights the importance of regular cardiovascular monitoring for CIRD patients, especially given their increased risk of more severe CAD. These findings support the need for longitudinal studies to explore the progression of CAD in CIRD patients over time, assessing how early intervention and changes in inflammatory markers correlate with improvements or worsening of cardiovascular health. Moreover, the involvement of specific coronary arteries (mid-anterior descending artery and right posterior descending artery) was higher in our CIRD patients. This could be important for understanding disease-specific patterns of atherosclerosis. Based on this, further studies could investigate the underlying mechanisms contributing to this preferential arterial involvement. In addition, revascularization was more common in our CIRD patients than in non-CIRD individuals, which could indicate that they experience more severe or widespread CAD, leading to higher rates of procedural intervention. Therefore, a follow-up study could examine the outcomes of revascularization in CIRD patients compared to non-CIRD patients.

  1. In some places, "CRID" is written instead of "CIRD".

Response: We apologize for the repeated mistake (nine times). The typographical error has now been corrected in all cases.

  1. As a reviewer, the most important question that arose after reading this study is: Should there be a standardized LDL-C target for initiating statin therapy in patients with CIRD, similar to the cardiovascular risk classification used for patients with diabetes? For instance, in individuals with diabetes alone, the LDL-C target is 70 mg/dL, while in those with established cardiovascular disease, it is 55 mg/dL. Is there an equivalent target value for patients with CIRD, or should there be one? What does the existing literature suggest on this matter, and what are the authors' recommendations? Please discuss this in the Discussion section.

Response: We greatly appreciate these accurate comments. In this regard, the Reviewer raises an important issue that we have incorporated into the Revised manuscript (Section of Discussion).

The EULAR (European League Against Rheumatism) 2015/2016 recommendations update guidance on managing CVD risk in patients with rheumatoid arthritis   and other CIRD emphasized the need of early detection and intervention, as these patients have a higher risk of CVD. In addition to using traditional CVD risk scores (such as SCORE and Framingham), which should be adjusted by a 1.5 multiplication factor in RA due to risk underestimation, the recommendations highlighted the importance of smoking cessation, weight control, and promoting physical activity. According to these recommendations, hypertension, diabetes, and lipid levels should be managed similarly to the general population. Furthermore, the recommendations emphasize the need for controlling systemic inflammation using disease-modifying antirheumatic drugs such as methotrexate, tumor necrosis factor   inhibitors, and interleukin-6   blockers to reduce cardiovascular risk [43].

Regarding lipid management, these recommendations indicate that statins should be considered for patients at high CVD risk, with treatment targets similar to those in the general population [43]. However, an unanswered question remains: should there be a standardized LDL-cholesterol target for initiating statin therapy in patients with CIRD, similar to the target used in the cardiovascular risk classification for patients with diabetes? In individuals with diabetes alone, the LDL-cholesterol target is 70 mg/dL, while in those with established cardiovascular disease, it is 55 mg/dL. Notably, many patients with CIRD have carotid plaques when assessed via carotid ultrasound. This finding places these patients in the category of very high cardiovascular risk. According to the 2019 ESC/EAS guidelines for the management of dyslipidemia, patients at very high cardiovascular risk should aim for an LDL-cholesterol reduction of ≥50% from baseline, with an LDL-cholesterol goal of <55 mg/d [44,45]. Moreover, since CIRD itself is considered a condition associated with high cardiovascular risk, as indicated in the 2019 ESC/EAS guidelines for the management of dyslipidemia [44,45], we feel that these patients should be treated with an LDL-cholesterol goal of <70 mg/dL.

Reviewer 2 Report

Comments and Suggestions for Authors

The study is well designed. I have two minor suggestions.

  1. Is it possible to add a forest plot?
  2. The discussion need to be strengthened. The rheumatoid diseases can cause microvascular dysfunction (Refer: 10.1093/eurheartj/ehw018) which is a major etiology of myocardial ischemia (Refer: 10.1631/jzus.B2100425). Some in-depth discussion on this point is essential. Also, please add the results on microvascular function evaluation if possible. 
  3. There are some errors in spelling and grammar, e.g., "rheumatoidd arthritis". Please double check thoroughly.
Comments on the Quality of English Language

Overall it is well written while further improvement is needed. 

Author Response

Comments and Suggestions for Authors

The study is well designed. I have two minor suggestions.

  1. Is it possible to add a forest plot?

Response: As indicated by the reviewer, in the Revised manuscript a forest plot is included below Table 3

Figure 1. Forest plot displaying odds ratios and 95% confidence intervals for each artery studied in Tables 2 and 3.

  1. The discussion need to be strengthened. The rheumatoid diseases can cause microvascular dysfunction (Refer: 10.1093/eurheartj/ehw018) which is a major etiology of myocardial ischemia (Refer: 10.1631/jzus.B2100425). Some in-depth discussion on this point is essential. Also, please add the results on microvascular function evaluation if possible. 

Response: We have incorporated these interesting references shown below and addressed the relevance of coronary microvascular dysfunction in patients with CIRD.

  1. 41. Faccini A, Kaski JC, Camici PG. Coronary microvascular dysfunction in chronic inflammatory rheumatoid diseases. Eur Heart J. 2016 Jun 14;37(23):1799-806. doi: 10.1093/eurheartj/ehw018. Epub 2016 Feb 23. PMID: 26912605.
  2. 42. Geng Y, Wu X, Liu H, Zheng D, Xia L. Index of microcirculatory resistance: state-of-the-art and potential applications in computational simulation of coronary artery disease. J Zhejiang Univ Sci B. 2022 Feb 15;23(2):123-140. English. doi: 10.1631/jzus.B2100425. PMID: 35187886; PMCID: PMC8861561.

We have included in the Discussion section the following paragraph:

An aspect not evaluated in our study was the implication of coronary microvascular dysfunction (CMD) in patients with CIRD. In this regard, Faccini et al. assessed the relationship between CIRD and CMD [41]. The authors highlighted how systemic inflammation in conditions such as rheumatoid arthritis and systemic lupus erythematosus contributes to CMD, independent of traditional cardiovascular risk factors. According to these authors, underlying mechanism, including endothelial dysfunction, immune system activation, and oxidative stress, lead to impaired coronary microcirculation. They emphasized the clinical relevance of CMD in these patients, as it increases the risk of cardiovascular events and warrants early detection and management [41]. Interestingly, more recently, Geng et al. highlighted the role of the Index of Microcirculatory Resistance (IMR), a key parameter for assessing coronary microvascular function [42]. The authors discussed how IMR can aid in identifying microvascular dysfunction, predicting patient outcomes, and guiding treatment strategies. This procedure may be applicable to patients with CIRD [42].

  1. There are some errors in spelling and grammar, e.g., "rheumatoidd arthritis". Please double check thoroughly.

Response: The typographical mistake has now been corrected. It was also the case for CIRD.

Round 2

Reviewer 1 Report

Comments and Suggestions for Authors

I have reviewed the revised manuscript titled "Identification of Coronary Morphological Damage in Patients with Chronic Inflammatory Rheumatic Diseases." The authors have addressed most of my comments effectively.

While it is clear that further long-term, large-scale mechanistic studies are needed to validate the results of this study and explore the underlying mechanisms, I believe that the findings of this work have the potential to make a valuable contribution to the literature and open doors for future research in this area.

Reviewer 2 Report

Comments and Suggestions for Authors

My earlier comments have been well addressed.

Comments on the Quality of English Language

Further polishing is needed.